# PD-L1 regulates inflammatory programs of macrophages from human pluripotent stem cells

Handi Cao[1],*, Yang Xiang[2],*, Shihui Zhang[1],*, Yiming Chao[1], Jilong Guo[2], Theo Aurich[2], Joshua WK Ho[2,3], Yuanhua Huang[1,2], Pentao Liu[1,2], Ryohichi Sugimura[1,2]

Programmed death ligand 1 (PD-L1) serves as a pivotal immune checkpoint in both the innate and adaptive immune systems. PD-L1 is expressed in macrophages in response to IFNγ. We examined whether PD-L1 might regulate macrophage development. We established PD-L1 KO (*CD274*$^{-/-}$) human pluripotent stem cells and differentiated them into macrophages and observed a 60% reduction in CD11B$^+$CD45$^+$ macrophages in *CD274*$^{-/-}$; this was orthogonally verified, with the PD-L1 inhibitor BMS-1166 reducing macrophages to the same fold. Single-cell RNA sequencing further confirmed the down-regulation of the macrophage-defining transcription factors *SPI1* and *MAFB*. Furthermore, *CD274*$^{-/-}$ macrophages reduced the level of inflammatory signals such as NF-κB and TNF, and chemokine secretion of the CXCL and CCL families. Anti-inflammatory TGF-β was up-regulated. Finally, we identified that *CD274*$^{-/-}$ macrophages significantly down-regulated interferon-stimulated genes despite the presence of IFNγ in the differentiation media. These data suggest that PD-L1 regulates inflammatory programs of macrophages from human pluripotent stem cells.

## Introduction

Programmed death ligand 1 (PD-L1) is abundant in tumors and suppresses T cells (1, 2). PD-L1, expressed in either cancer cells or myeloid cells, binds with its receptor PD-1 in T cells (3). PD-L1 is one of the major targets of cancer immunotherapy (4). Deletion of the PD-L1 encoding gene *CD274* in mouse models provoked T-cell activation (5). However, whether PD-L1 regulates other immune cell types such as macrophages is less known. PD-L1 expression is induced by both IFNγ and TLR agonists (6, 7, 8).

Macrophages are a versatile cell source for immunotherapy (9, 10, 11), and their fine-tuning of inflammatory programs is the key to successful cell therapy against cancer and fibrosis (12). Particularly,

macrophages from human pluripotent stem cells (hPSCs) are expected to serve as a curative source of cancer immunotherapy (13). Understanding whether and how PD-L1 might regulate macrophages is an important question. Here, we examined the role of PD-L1 in macrophages derived from hPSCs. Both genetic and pharmacological inhibition of PD-L1 impaired macrophage development. The resultant macrophages deflect the inflammatory program upon IFNγ treatment. These observations indicate that PD-L1 regulates inflammatory programs of macrophages from hPSCs.

## Results

We determined the role of PD-L1 in macrophage development from hPSCs to yolk sac organoids (Fig 1A and B). We formed EBs (embryoid bodies) and applied STEMdiff Hematopoietic Kit for mesoderm patterning and differentiation into yolk sac organoids (14). We induced them into monocytes with STEMdiff monocyte differentiation supplement and then into subsequent maturation to macrophages with the ImmunoCult-SF macrophage medium. We defined macrophages by the expression of CD11B, CD14, CD16, CD45, and CD68 (Figs 1C and S1). We polarized macrophages to proinflammatory status with LPS and IFNγ, whereas they were polarized to anti-inflammatory status by IL-4. PD-L1 was predominantly expressed in proinflammatory macrophages. In contrast, the expression of its receptor PD-1 could not be detected during the differentiation of yolk sac organoids (Fig 1C). Moreover, PD-L1 was expressed exclusively upon exposure to IFNγ (Fig 1D).

To examine the role of PD-L1 in macrophage development, we established *CD274*$^{-/-}$ hPSCs by targeting exon 2 of the *PD-L1* gene, which encodes PD-L1 (Fig S2A, B, and D), and *CD274*$^{-/-}$ hPSCs still expressed the pluripotency markers (SSEA3, SSEA4, Tra-1-60, and Tra-1-81) similar to WT hPSCs (Fig S2C). We confirmed the KO efficiency (Figs 2A and B and S2E and F). The percentage of CD45$^+$CD11B$^+$ macrophages was significantly decreased in *CD274*$^{-/-}$ yolk sac organoids, suggesting that PD-L1 regulates macrophage development (Fig 2C and D). Two independent *CD274*$^{-/-}$ hPSC lines,

[1]Centre for Translational Stem Cell Biology, Hong Kong, China   [2]School of Biomedical Sciences, Li Ka Shing Faculty of Medicine, The University of Hong Kong, Hong Kong, China   [3]Laboratory of Data Discovery for Health Limited (D24H), Hong Kong Science Park, Hong Kong, China

Correspondence: rios@hku.hk
*Handi Cao, Yang Xiang, and Shihui Zhang contributed equally to this work

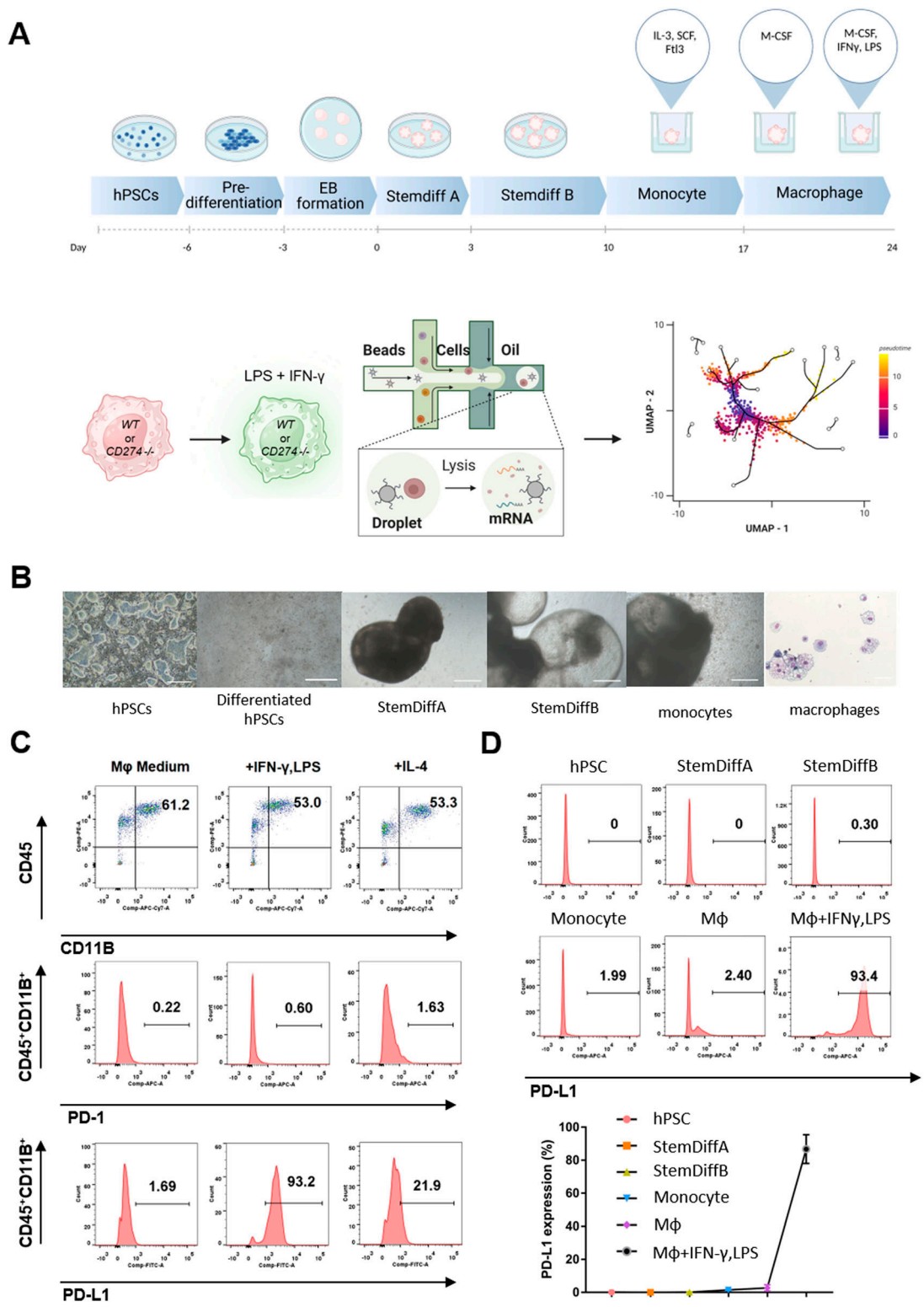

**Figure 1. Human macrophages from yolk sac organoids express PD-L1 upon IFNγ stimulation.**
**(A)** Schematic overview of the generation of human yolk sac organoids and macrophages. Human pluripotent stem cells (hPSCs) were differentiated into yolk sac organoids by commercial media (STEMdiff), and then, macrophages were induced with CSF-1 (the upper graph). The *CD274*[+/+] or *CD274*[−/−] hPSCs were differentiated into macrophages treated with IFNγ and LPS, and then, the stimulated macrophages were used in the following single-cell RNA sequencing (the bottom graph). **(B)** Bright-field images of representative cellular morphology from hPSC differentiation into human yolk sac organoids and macrophages. EB: embryoid body; HPC: hematopoietic progenitor cell. **(C)** Flow cytometry analysis of CD45, CD11B, PD-1, and PD-L1 on macrophages in macrophage basal medium (Mφ medium), Mφ medium plus IFNγ and LPS, and Mφ medium plus

A7-1 and B2-3, showed similar reductions in macrophages (Fig S3). The PD-L1 inhibitor BMS-1166 (15) phenocopied the KO lines by reducing the percentage of CD45$^+$CD11B$^+$ macrophages (Fig 2E). These data suggest that PD-L1 supports the development of macrophages in yolk sac organoids.

To further characterize $CD274^{-/-}$ macrophages, we performed 10X Chromium single-cell RNA sequencing (scRNA-seq). We sequenced whole live cells from yolk sac organoids after induction to proinflammatory macrophages. We subsequently removed low-quality cells (Fig S4A) and doublets with DoubletFinder (Fig S4B). We obtained a total of 25,439 cells of which 16,258 were *WT* and 9,181 were $CD274^{-/-}$ yolk sac organoids. We annotated 19 cell types according to canonical markers and visualized them with UMAP (Fig S4C and E). We defined the monocyte and macrophage (MC/Mφ) population expressing *CD14*, *CD68*, *CD163*, and *FCGR3A* (encodes CD16) (Figs 2F and S4C and D). We consistently detected a significant reduction in macrophages using both flow cytometry and scRNA-seq (Fig 2D versus Fig 2G).

We next determined the differentially expressed genes (DEGs) in MC/Mφ between *WT* and $CD274^{-/-}$ yolk sac organoids (Figs 3A and S5). $CD274^{-/-}$ MC/Mφs exhibited the reduced expression of the macrophage-determining transcription factors (TFs) *SPI1*, *KLF6*, and *MAFB* (Figs 3B and 4C), which was consistent with PD-L1 inhibitor BMS-1166–treated MC/Mφs (Fig 4F). Gene Ontology (GO) and KEGG analyses by GSEA revealed significant down-regulation of proinflammatory-associated signaling pathways such as NOD, chemokine, TNF, NF-κB, and IL-17, and antigen presentation signaling pathways and up-regulation of an anti-inflammatory TGF-β signaling pathway (Fig 3C). This suggests that PD-L1 may regulate inflammatory macrophage development.

Consistently, cell–cell communication analysis showed a reduced macrophage activation ligand–receptor network in $CD274^{-/-}$ MC/Mφs such as IL-6, IL-10, CSF, and chemokines CCL and CXCL. However, $CD274^{-/-}$ MC/Mφs increased the anti-inflammatory TGF-β ligand–receptor network (Fig 3D). We validated the reduction in chemokine secretion from the $CD274^{-/-}$ MC/Mφ (Fig 4A and B) and the PD-L1 inhibitor BMS-1166–treated MC/Mφ (Fig 4D and E). We identified increased GDF and THBS signals. GDF15, also called macrophage inhibitory cytokine-1 (16), significantly interacts with TGFBR2. THBS1 interacted with the macrophage suppressor CD47 (17) (Fig S6). These data indicate that $CD274^{-/-}$ MC/Mφs were anti-inflammatory.

We then examined whether $CD274^{-/-}$ MC/Mφs respond to IFNγ present in the macrophage activation media (Fig 1A). GSEA revealed that interferon-stimulated genes (ISGs) were significantly down-regulated in $CD274^{-/-}$ MC/Mφs, suggesting that these cells could not respond to IFNγ (Fig 3E). $CD274^{-/-}$ MC/Mφs exhibited the reduced expression of the interferon receptors IFNGR1 and IFNAR1 (Figs 3F and 4C), which was orthogonally validated in PD-L1 inhibitor BMS-1166–treated MC/Mφs (Fig 4F). These results demonstrate that PD-L1 may maintain IFNγ signaling by sustaining the expression of its receptor IFNGR1 and establishing inflammatory programs.

Finally, we examined the resemblance of human macrophages between yolk sac organoids and fetal yolk sacs (18). We observed that cellular components such as macrophages (MC/Mφs), common myeloid progenitors, granulocytes, and the yolk sac myeloid-biased progenitors were correlated between yolk sac organoids (labeled as PD-L1 WT) and fetal yolk sacs (labeled as in vivo YS) (Fig S7). Taken together, our data indicate that PD-L1 regulates the development of inflammatory macrophages in human yolk sacs. Mechanistically, PD-L1 maintains IFNγ signaling to establish inflammatory programs in macrophages.

## Discussion

PD-L1 inhibition reduced inflammatory macrophages, as well as ISG and IFNGR1 expression, resulting in $CD274^{-/-}$ macrophages that were unresponsive to IFNγ. Our data suggest that PD-L1 maintains the integrity of the IFNγ signal by sustaining the expression of IFNGR1. How and whether PD-L1 directly regulates ISG and IFNGR1 expression are intriguing questions. The reduction in macrophage-determining TFs by *CD274* KO implies that PD-L1 governs the gene regulatory network of macrophage inflammatory programs. Nuclear PD-L1 might be a potential mechanism to maintain IFNγ signaling and the macrophage gene regulatory network (19, 20).

The use of hPSCs offered us the advantage of studying human macrophage development in the yolk sac. We previously established yolk sac organoids from hPSCs that commit to erythro/megakaryocyte and monocyte lineages (14). In this study, we employed M-CSF to induce macrophages in yolk sac organoids. The macrophages in our study resemble those of fetal-stage yolk sacs, as indicated by cross-referencing of scRNA-seq datasets (18). This study proposed the potentially novel role of PD-L1 in regulating inflammatory programs of macrophages, as well as utility of hPSCs in understanding the biology of human immune cells.

## Materials and Methods

### hPSC maintenance and EB formation

hPSCs were obtained from the Pentao Liu laboratory and maintained in a medium with chemically defined conditions (21). The KSR medium was removed, and the cells were washed with PBS. The cells were digested with TrypLE (12605036; Gibco) at 37°C for ~5 min. TrypLE was removed, and the KSR medium was added. Cells were harvested, avoiding harsh pipetting. The cells were centrifuged at 300$g$ for 3 min. The supernatant was carefully removed. The KSR medium and the ROCK inhibitor Y-27632 (10 μM) were added to resuspend the cells. Each 30 μl hanging drop containing 5,000 cells was placed onto the inside of the lid of a 100-mm petri dish (20100; SPL). ~30–40 drops were placed on each lid, and the dish was filled

---

IL-4. **(D)** Flow cytometry analysis of PD-L1 expression in hPSCs, human yolk sac organoids (STEMdiff A and STEMdiff B), and macrophages. Representative flow cytometry data are shown here. Data are shown as the mean ± SEM. N = 5 measurements from two independent experiments performed with 2~3 technical replicates.

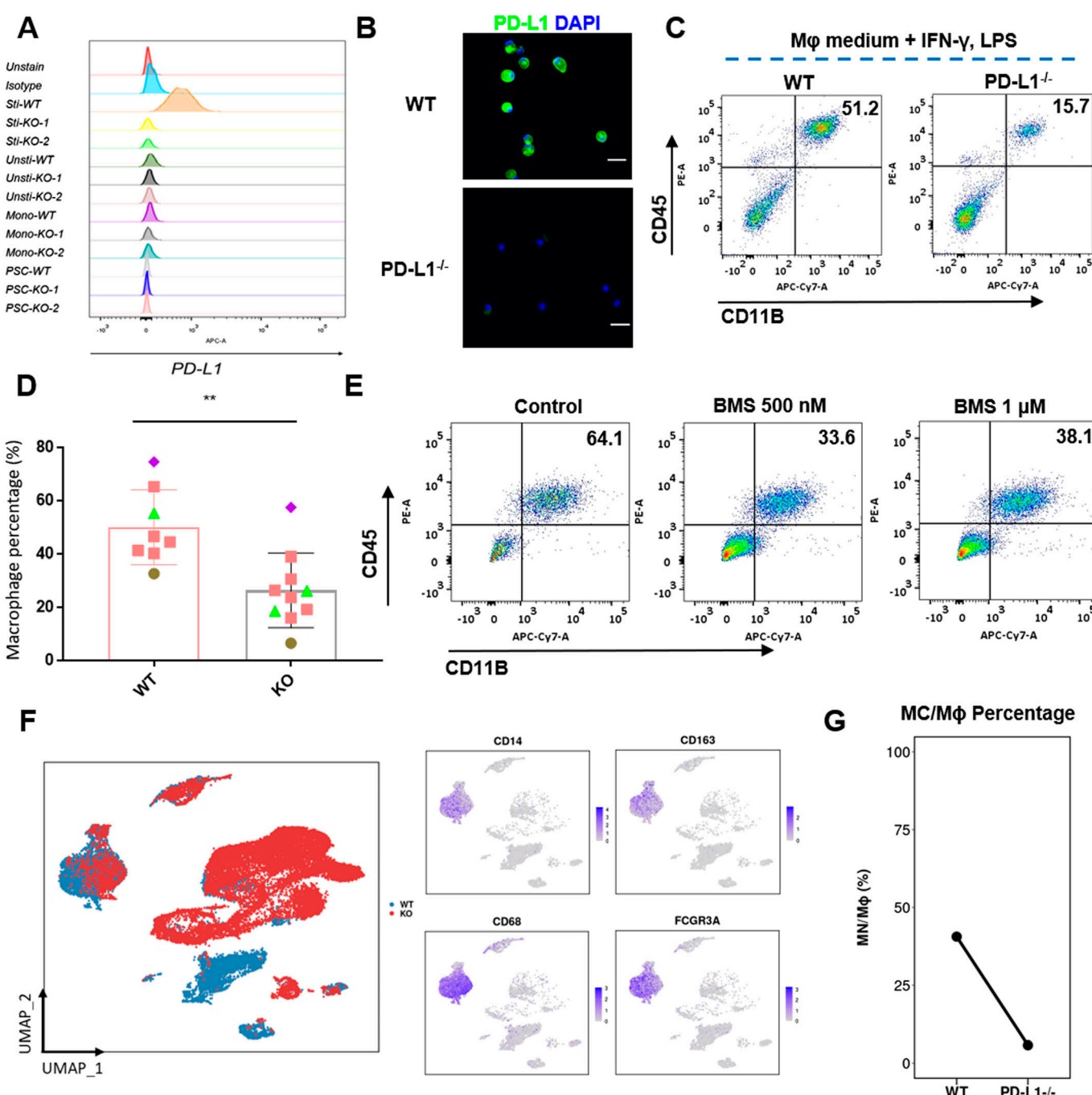

**Figure 2. KO of *PD-L1* prevents macrophage development.**
**(A)** Flow cytometry analysis of PD-L1 expression of the WT and two PD-L1 knockouts at different stages (from PSCs, monocytes, unstimulated macrophages to stimulated macrophages). IgG isotype is used as a control for gate-positive population. Unsti: unstimulated; Sti: stimulated. **(B)** Immunofluorescence of PD-L1 expression on *WT* and *CD274⁻/⁻* macrophages in Mφ medium plus IFNγ and LPS. Scale bar: 50 μm. **(C, D)** Flow cytometry analysis of macrophage (CD45⁺CD11B⁺) (C) and percentage (D) of WT and *CD274⁻/⁻* macrophages in Mφ medium plus IFNγ and LPS. Statistical results shown here were from five independent experiments (n = 8–9). **P < 0.01. **(E)** PD-L1 inhibitor BMS-1166 reduced macrophage development in human yolk sac organoids. Flow cytometry analysis of CD45 and CD11B on macrophages after 500 nM or 1 μM BMS-1166 treatment (below). Representative flow cytometry data are shown here. **(F)** Uniform Manifold Approximation and Projection visualization of single cells from yolk sac organoids in WT (n = 9,181, in blue) and *PD-L1* KO (n = 16,258, in red). Each dot represents a single cell. In total, 19 cell types were annotated and are displayed in different colors (see Fig S5). The right panel shows the expression of macrophages and monocyte-specific marker genes. **(G)** Dot-line chart displaying the decrease in monocyte/macrophage (MC/Mφ) proportion upon *PD-L1* KO. Statistical significance (*P*-value) was calculated by the likelihood-ratio test and adjusted by the Benjamini–Hochberg method using the R package DCATS.

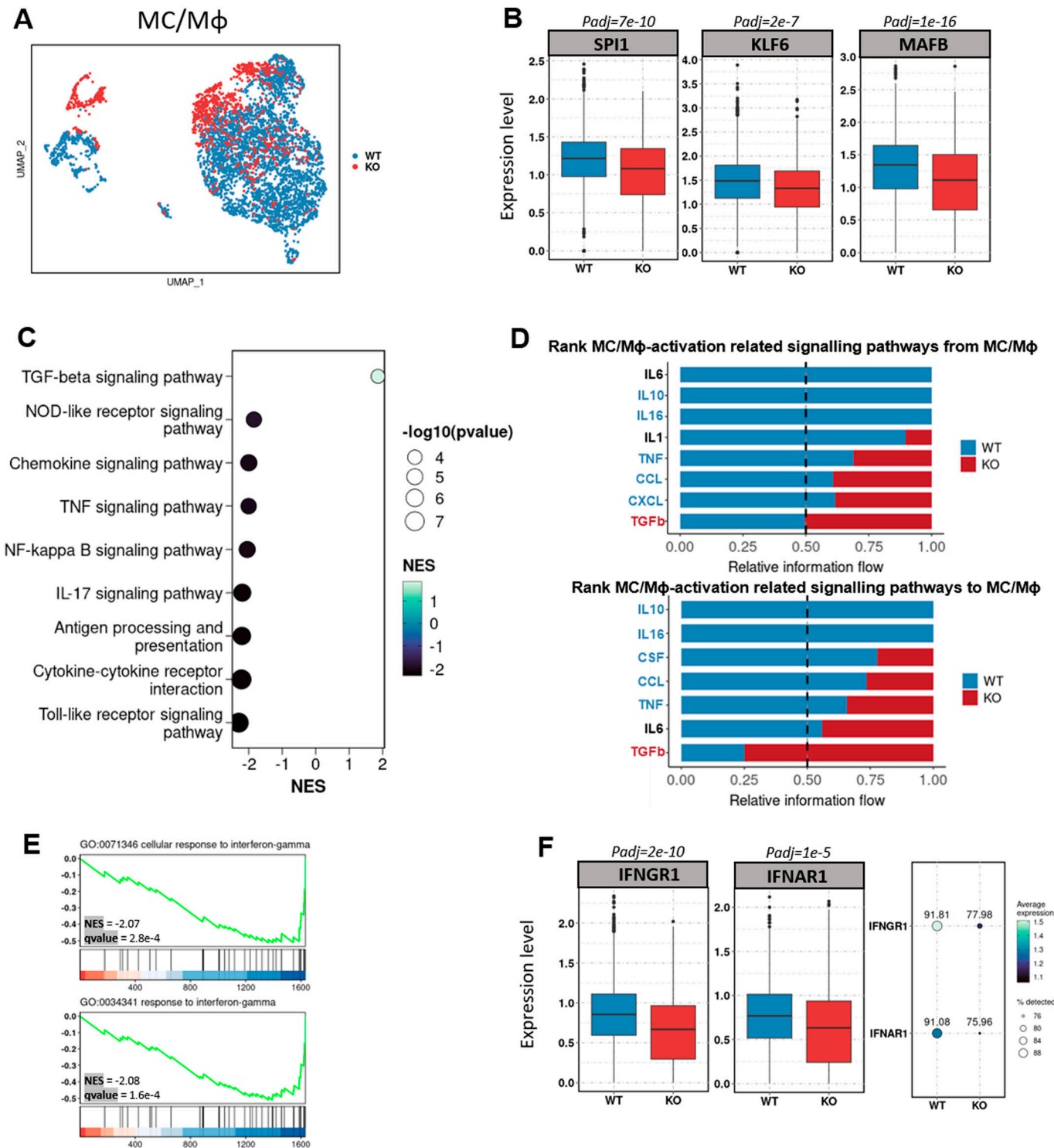

**Figure 3. CD274$^{-/-}$ macrophages down-regulate inflammatory programs and are unable to respond to IFNγ.**
**(A)** UMAP visualization of monocyte and macrophage (MC/Mϕ) populations colored by genotype (WT, PD-L1 KO). **(B)** Box plot showing the decreased expression of selected macrophage-determining transcription factors within the MC/Mϕ cluster upon PD-L1 KO. (P-value was calculated using the "bimod" test in the Seurat R package and corrected by the "Benjamini–Hochberg" method.) **(C)** Selected KEGG pathways enriched by gene set enrichment analysis of differentially expressed genes between PD-L1 KO and WT cells within the MC/Mϕ cluster. NES, normalized enrichment score. Differentially expressed genes were calculated using the same method described in Fig 3B. **(D)** Bar graphs showing the ranking of major outgoing (upper panel) and incoming (lower panel) signals of MC/Mϕ upon PD-L1 KO compared with WT. The rank of signals was based on differences in the overall information flow of each group. **(E)** Gene set enrichment analysis shows the enrichment of differentially expressed down-regulated genes upon PD-L1 KO in MC/Mϕ cells in the interferon-gamma response. **(F)** Box plot and dot plot showing the reduction in both the expression and percentage of cells expressing interferon-

with PBS to maintain moisture. The lid was inverted gently to cover the dish. The dishes with EBs were incubated for 3 d.

### Generation of macrophages from hPSC-derived yolk sac organoids

All EBs were collected with a 1,000-µl pipette and centrifuged at 20g for 1 min. The supernatant was carefully removed. EBs were cultured in an ultra-low-adherence plate with STEMdiff A (STEMdiff Hematopoietic Kit; STEMCELL Technologies) for 3 d and STEMdiff B (STEMdiff Hematopoietic Kit; STEMCELL Technologies) for 7 d as described previously (14). All cell clumps formed in STEMdiff B were then transferred to a 24-well Transwell plate (0.4 µM pore, catalog no. 38024; STEMCELL Technologies) for further differentiation. We added three cytokines into the monocyte medium (catalog no. 05320; STEMCELL Technologies), including 5 ng/ml IL-3 (catalog no. 203-IL-010; R&D Systems), 20 ng/ml SCF (catalog no. 255-SC-010; R&D Systems), and 10 ng/ml FLT3L (catalog no. 308-FK-005; R&D Systems), and cultured organoids for 7 d. The medium was changed every other day. And then, we cultured organoids with Macrophage Differentiation Medium (catalog no. 10961; STEMCELL Technologies) supplemented with 50 ng/ml human CSF-1(catalog no. 216-MC-100; R&D Systems). Four days later, we obtained macrophages, and then, we added either human 50 ng/ml CSF-1, 50 ng/ml human IFN-gamma (catalog no. 285-IF-100; R&D Systems), and 10 ng/ml LPS (catalog no. L2630-100MG; Sigma-Aldrich), or human 50 ng/ml CSF-1 and 10 ng/ml IL-4 (catalog no. 204-IL-010; R&D Systems) into the well. In 2 d, we harvested adhering IFN-gamma– + LPS- + CSF-1– stimulated proinflammatory macrophages or IL-4– + CSF-1– stimulated anti-inflammatory macrophages.

### Chemical inhibition of PD-L1 by BMS-1166

For the drug treatment experiments, we generated hPSC-derived organoids as mentioned above. When EB grew up at the STEMdiff B medium (STEMdiff Hematopoietic Kit; STEMCELL Technologies) for 7 d, we added a chemical drug BMS-1166 (catalog no. S8859; Selleck Chemicals) at the concentration of 500 nM. The chemical drug would be provided from the monocyte stage to the macrophage stage around two weeks.

### Establishment of *CD274*$^{-/-}$ hPSCs with CRISPR/Cas9

sgRNAs were designed to target *PD-L1/PD-L1* exon 2. The designs are described at https://chopchop.cbu.uib.no/. sgRNAs were synthesized by IDT technologies. The editing efficiency of these sgRNAs was initially validated in hPSCs by electroporation. The highly edited sgRNA sequence is shown in Fig S2. CRISPR/Cas9 KO was conducted as follows. The synthetic oligos were annealed, and then, the annealed double-strand oligos and backbone vectors were digested by the restriction enzyme BbsI (catalog no. ER1011; Thermo Fisher Scientific). After enzyme digestion, the products were purified. Purified oligos and vectors were ligated overnight. On the

next day, bacterial transformation was performed for recombinant PD-L1 KO construct amplification. The extracted plasmid sequences were then confirmed by Sanger sequencing. *PD-L1 KO* piggyBac plasmids were generously provided by the Pentao Liu laboratory. hPSCs were maintained for 4–5 d before electroporation. The confluency was approximately 70~80%. hPSCs were electroporated using Human Stem Cell Nucleofector Kit 2 (catalog no. LONVPH-5022; Lonza). After electroporation, hPSCs were reseeded on the plate, and the medium was changed the next day. Three days later, cells were selected by the addition of 2 µg/ml puromycin (catalog no. P9620-10ML; Sigma-Aldrich). After several days of selection, the colonies were picked, expanded, and genotyped. The two pairs of genotyping primers were designed as follows: PD-L1-1F, AACC-GACCAGATAAAGTGATT and PD-L1-1R, ATCCTGCAATTTCACATCTGTGA; and PD-L1-2F, ATAAACGCTGTGCCAATTTTGT and PD-L1-2R, TCATG-CAGCGGTACACCCCTG. Single colonies confirmed to have inherited the deletion by Sanger sequencing were selected for the following experiments.

### Flow cytometry analysis

Cell suspensions were prepared for flow cytometry analysis. For the hPSC line, TrypLE was used for dissociation. For EBs and other cell clumps from STEMdiff A and B, monocyte induction medium, and macrophage induction medium, ACCUMAX (catalog number) was used for dissociation. To collect attached cells in Transwells, ACCUMAX was also used. The cells were resuspended in 100 µl FACS staining buffer, and 5 µl FcX (human, 422302; BioLegend) was added to each sample at room temperature for 10 min. Stain cell surface markers with optimal concentrations of specific antibodies (CD11B-APC-Cy7, CD45-PE, CD14-FITC, CD16-PE-Cy7, CD68-PE-Cy7, PD-1-FITC, and PD-L1-APC) were used for extracellular staining for 30 min on ice following a previous panel design. The cells were washed once with PBS. The samples were centrifuged at 400–600g at RT for 5 min, and the supernatant was discarded. DAPI was added to all samples before analysis. Analysis was performed on a BD FACSymphony A3 cell analyzer. For all analyses, single cells were gated based on FSC-A versus FSC-H, and DAPI+ dead cells were gated out.

### May–Grunwald–Giemsa staining

All cells after the macrophage stage were collected and used for cytospin on glass slides with a program of 60g for 3 min. After drying slides overnight, they were immersed in 100% May–Grunwald stain for 4 min followed by washing and then were transferred to 4% Giemsa stain for another 4 min. The slides were rinsed in tap water to ensure that the background blue stain was removed. Slides were dried before taking photographs under a microscope.

### Immunofluorescence

WT or KO total cells were respectively fixed in 4% formaldehyde (catalog no. P0099-100ML; Beyotime) for 20 min at room

---

gamma receptor *IFNGR1* and interferon-alpha receptor *IFNAR1*, respectively, within the MC/Mφ cluster upon *PD-L1* KO. (*P*-value was calculated using the "bimod" test in the Seurat R package and corrected by the "Benjamini–Hochberg" method).

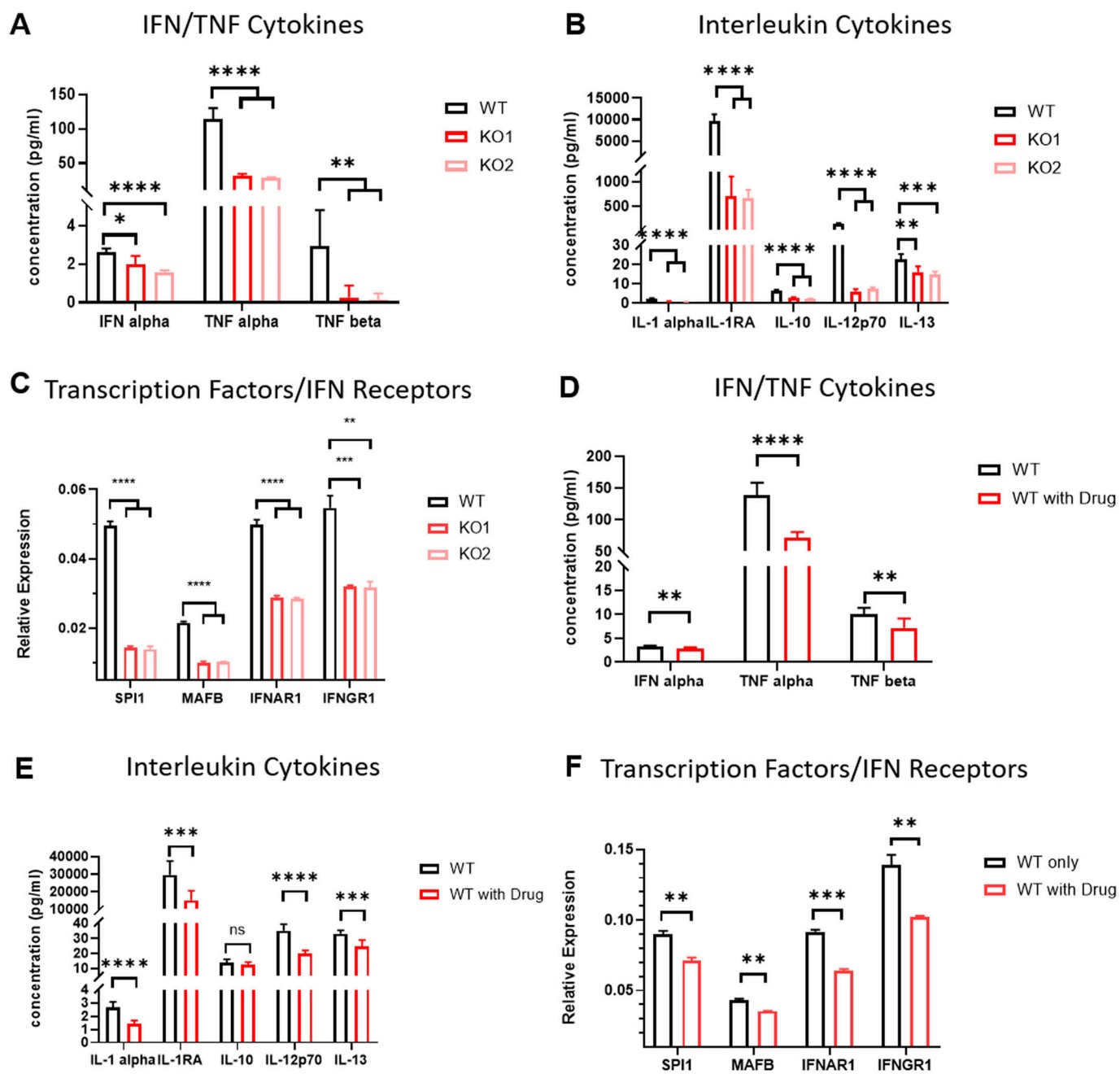

**Figure 4.** *PD-L1 genetic knockout and pharmacologically inhibited* macrophages down-regulate inflammatory programs and macrophage-related transcription factors.

**(A)** Luminex profiling of secreted IFN/TNF from MC/Mϕs upon *PD-L1* KO compared with WT (WT, *PD-L1* KO). Data are shown as the mean ± SEM. N = 3~4 measurements from one independent experiment performed with two technical replicates.*P < 0.05, **P < 0.01, ***P < 0.001, and ****P < 0.0001. **(B)** Luminex profiling of secreted interleukins from MC/Mϕ upon *PD-L1* KO compared with WT (WT, *PD-L1* KO). Data are shown as the mean ± SEM. N = 3~4 measurements from one independent experiment performed with two technical replicates. *P < 0.05, **P < 0.01, ***P < 0.001, and ****P < 0.0001. **(C)** qRT-PCR analysis of transcription factors (SPI1/MAFB) and interferon receptors (IFNAR1/IFNGR1) between WT and *PD-L1* KO. The relative expression is normalized by the GAPDH expression level. Data are shown as the mean ± SEM. N = 3~4 measurements from one independent experiment performed with two technical replicates. **P < 0.01, ***P < 0.001, and ****P < 0.0001. **(D)** Luminex profiling of interferon-related cytokines (IFN/TNF) from MC/Mϕs upon the PD-L1 chemical inhibitor group (WT with drug) compared with WT. Data are shown as the mean ± SEM. N = 3~4 measurements from one independent experiment performed with two technical replicates.*P < 0.05, **P < 0.01, ***P < 0.001, ****P < 0.0001, and ns: no significance. **(E)** Luminex profiling of interleukins (IL-1/IL-10/IL-12/IL-13) from MC/Mϕs upon the PD-L1 chemical inhibitor group (WT with drug) compared with WT. Data are shown as the mean ± SEM. N = 3~4 measurements from one independent experiment performed with two technical replicates.*P < 0.05, **P < 0.01, ***P < 0.001, ****P < 0.0001, and ns: no significance. **(F)** qRT-PCR analysis of transcription factors (SPI1/MAFB) and interferon receptors (IFNAR1/IFNGR1) between WT and the PD-L1 chemical inhibitor group (WT with drug). The relative expression is normalized by the GAPDH expression level. Data are shown as the mean ± SEM. N = 3~4 measurements from one independent experiment performed with two technical replicates. **P < 0.01 and ***P < 0.001.

Source data are available for this figure.

temperature, then the cells were spun down, the supernatant was poured off, and the cell pellets were resuspended in 200 $\mu$l cell suspension buffer. After cytospin, the slides were permeabilized with 0.5% Triton X-100 (catalog no. 93443-100ML; Sigma-Aldrich) for 30 min and blocked with 10% BSA solution (catalog no. 126615-25ML; MilliporeSigma) for 1 h. The slides were washed twice and then stained with the primary antibody anti-PD-L1 (catalog no. ab213524; Abcam) overnight in a wet box. After 12 h, the slides were rinsed thrice with PBS, and then, the secondary antibody Goat Anti-Rabbit IgG H&L (Alexa Fluor 488) (catalog no. ab150077; Abcam) was added and incubated for 1 h. The slides were rinsed thrice with PBS, and 100 $\mu$l diluted DAPI buffer (catalog no. 564907; BD Bioscience) was added to each slide and incubated for 3 min at room temperature. The slides were washed twice with PBS and then kept wet. Images were taken by a Nikon Ti2E fluorescence microscope and merged using ImageJ software.

## Cytokine detection by Luminex

On day 0, 200 k macrophages, which were from the stimulated macrophage stage, were seeded. And then, cells were cultured for 5 d. Then, supernatants were harvested, centrifuged at 184$g$ for 10 min, kept for the following experiments, and snap-frozen in the −80°C deep freezer. The wash buffer was used for bead cleaning. A standard mix was used for the generation of standard curves. The reading buffer was used for sample running on the machine.

A 96-well plate was prepared, and beads from Cytokine & Chemokine Convenience 34-Plex Human ProcartaPlex Panel 1A kits (#EPXR340-12167-901) were added to the plate. The bead coating plate was put on the magnetic plate to settle down the beads. The liquid was removed. The samples and standard samples were added to the corresponding wells on the plate. The plate was sealed and shaken at 600 rpm for 2 h at 25°C. After the cytokines bound to the beads, the seal was removed and the plate was washed with washing buffer thrice by the magnetic plate. Then, 25 $\mu$l biotinylated detection antibody mix was added to each well on the plate, and then, the plate was shaken at 600 rpm for 30 min at 25°C. The washing steps were repeated as before. Afterward, 50 $\mu$l streptavidin–PE was added to the plate in order to bind to the detection antibody. The washing steps were repeated, then 120 $\mu$l reading buffer was added to each well, and the plate was shaken at 34$g$ for 10 min at 25°C. Finally, the plate was run on the Luminex xMAP instrument.

## Single-cell RNA sequencing

### Sample preparation

The WT and $CD274^{-/-}$ hPSC lines followed macrophage differentiation. All cells were collected and resuspended in FACS buffer. Cells were stained with CD11b-APC-Cy7 and CD45-PE. After staining, the cells were washed and resuspended in FACS buffer with DAPI. Cells were sorted using Fusion Flow Cytometer (BD Bioscience). DAPI-negative cells were sorted for single-cell RNA-seq with the support of the Centre for PanorOmic Sciences. Briefly, the single-cell RNA-seq library was constructed by Chromium Next GEM Single Cell 3′ Reagent Kit, v3.1. An Illumina NovaSeq 6000 was used for paired-end sequencing services, and the final data output was 206.2 G for $CD274^{-/-}$ and 213.9 G for WT samples.

### Data analysis

Single-cell transcriptomic sequencing data of WT and $CD274^{-/-}$ samples from 10X Genomics were processed with CellRanger software (v7.0.0) and mapped to the GRCh38 human reference genome. The read counts of each gene by cells were counted and loaded with the Seurat R package (v4.2.0). Quality control and cell filtering were conducted for the two datasets separately. Cells with fewer than 500 detected features, 2,000 counts, or more than 15% mitochondrial gene expression were removed. Doublets detected by the Doublet Finder R package (v2.0.3) with either high or low confidence were subsequently filtered out.

After quality control, we checked the batch effect between our two datasets using the Seurat FindIntegrationAnchors function. The top 2,000 highly variable genes were used for principal component analysis to conduct dimension reduction and clustering and were further visualized by Uniform Manifold Approximation and Projection. Markers of each cluster were identified by the Seurat FindAllMarkers function, which together with well-known canonical gene markers were used for cell-type annotation.

DEG analysis of the macrophage population (MC/M$\varphi$) between WT and $CD274^{-/-}$ samples was performed using the Seurat Find-Markers function and the "bimod" test (likelihood-ratio test for single-cell gene expression). DEGs were defined as genes with a q-value < 0.05 and an absolute value of average log$_2$ fold change > 0.1.

Gene Ontology (GO) and KEGG analyses of all DEGs (all genes were kept but with a q-value equal to NA) were performed using gene set enrichment analysis and the gseGO and gseKEGG functions of the clusterProfiler R package (v4.4.4). The enriched GO and KEGG terms of interest, defined by a q-value < 0.05 and an absolute value of normalized enrichment score > 1, were visualized by a dot plot.

Cell–cell communication analysis was performed with the CellChat R package (v1.5.0). The filtered gene expression raw data of our two datasets with annotated cell-type information were loaded to establish the CellChat object. The ligand and receptors (L-R) within different cell populations were then recognized and calculated for L-R pair probability. To compare the cell interactions between WT and $CD274^{-/-}$ samples, we then merged these two CellChat objects into one object. The total number of interactions and strength of cell–cell communication networks were inferred using the compareInteractions function. The information flow of a specific signaling network was compared between WT and $CD274^{-/-}$ samples and then visualized with the rankNet function as a stacked bar plot with sources and targets set as the macrophage group (MC/M$\varphi$) separately.

The statistical significance analysis of differential cell-type abundance between WT and $CD274^{-/-}$ samples was performed using the DCATS R package [22], with default parameters. An LRT_pval ($P$-value calculated by the likelihood-ratio test) less than 0.01 was used to define whether a cell type showed differential proportions upon PD-L1 KO.

## Statistical analysis

All results were derived from at least three independent experiments. Values are expressed as the mean ± SD. Comparisons between groups were performed using an unpaired *t* test or one-way ANOVA (GraphPad Prism 7.00 Software, Inc.). *P*-values less than 0.05 were accepted to indicate statistically significant differences.

# Data Availability

The scRNA-seq data reported in this study have been deposited in NCBI with the accession number GSE218722.

# Supplementary Information

# Acknowledgements

We thank CPOS at HKUMed for technical assistance in scRNA-seq and FACS. We thank Cheryl Tam for operating the cell sorting. We thank American Journal Experts (AJE) for English proof service. We thank Heidi Ling and Alice Kwok for assisting the Luminex assay. We thank Danny Chan and Yuanhua Huang's lab members for discussion, and members of the Centre for Translational Stem Cell Biology for discussion and administrative support. The study was supported by the Platform for Technology Fund, RGC ECS 27109921; the Seed Fund from the School of Biomedical Sciences; and the InnoHK Centre for Translational Stem Cell Biology.

## Author Contributions

H Cao: conceptualization, resources, data curation, validation, investigation, visualization, methodology, project administration, and writing—original draft, review, and editing.
Y Xiang: conceptualization, resources, data curation, software, validation, investigation, visualization, methodology, project administration, and writing—original draft, review, and editing.
S Zhang: resources, data curation, software, formal analysis, visualization, project administration, and writing—original draft, review, and editing.
Y Chao: data curation, formal analysis, supervision, and visualization.
J Guo: conceptualization, resources, methodology, and project administration.
T Aurich: data curation, validation, visualization, methodology, and project administration.
JWK Ho: resources, data curation, software, formal analysis, supervision, visualization, and project administration.
Y Huang: resources, data curation, software, formal analysis, supervision, visualization, and project administration.
P Liu: resources, supervision, validation, investigation, visualization, methodology, and project administration.
R Sugimura: conceptualization, resources, data curation, software, formal analysis, supervision, funding acquisition, validation, investigation, visualization, methodology, project administration, and writing—original draft, review, and editing.

## Conflict of Interest Statement

The authors declare that they have no conflict of interest.

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
