## [Reviewer comments · Life Science Alliance]

Life Science Alliance

PD-L1 regulates inflammatory programs of macrophages from human pluripotent stem cells

Handi CAO, Yang XIANG, Shihui ZHANG, Yiming CHAO, Jilong GUO, Theo Aurich, Joshua W. K. Ho, Yuanhua Huang, Pentao Liu, and Ryohichi Sugimura

DOI: <https://doi.org/10.26508/lsa.202302461>

Corresponding author(s): Ryohichi Sugimura, University of Hong Kong

Review Timeline:

Submission Date:	2023-10-27
Editorial Decision:	2023-10-30
Revision Received:	2023-11-02
Accepted:	2023-11-03

Transaction Report:

Please note that the manuscript was previously reviewed at another journal and the reports were taken into account in the decision-making process at *Life Science Alliance*. Since the original reviews are not subject to Life Science Alliance's transparent review process policy, the reports and author response cannot be published.

October 30, 2023

RE: Life Science Alliance Manuscript #LSA-2023-02461

Dr. Ryohichi Sugimura
University of Hong Kong
L1-59, Lab Block, 21 Sassoon Rd, Pok Fu Lam
Hong Kong

Dear Dr. Sugimura,

Thank you for submitting your revised manuscript entitled "PD-L1 regulates inflammatory programs of macrophages from human pluripotent stem cells". We would be happy to publish your paper in Life Science Alliance pending final revisions necessary to meet our formatting guidelines.

- please upload all figure files as individual ones, including the supplementary figure files; all figure legends should only appear in the main manuscript file
- please upload your main manuscript text as an editable doc file
- please add ORCID ID for the corresponding (and secondary corresponding) author--you should have received instructions on how to do so
- please add a Summary Blurb/Alternate Abstract in our system (that is not a title)
- please add the Twitter handle of your host institute/organization as well as your own or/and one of the authors in our system
- please make sure the author order in your manuscript and our system match
- please use the [10 author names et al.] format in your references (i.e., limit the author names to the first 10)
- if possible, please provide one figure per page
- there is a callout for figure 3G in the manuscript text, and this figure does not have a panel G. -- please correct
- please add callouts for Figures 4A-F; S2A-F and S4A-E to your main manuscript text
- the contributions indicated for the following authors do not qualify them for authorship. Please either update their contributions, or let us know if they need to be removed as authors. Joshua W. K. Ho, Yuanhua Huang, and Pentao Liu

A. FINAL FILES:

B. MANUSCRIPT ORGANIZATION AND FORMATTING:

Sincerely,

November 3, 2023

RE: Life Science Alliance Manuscript #LSA-2023-02461R

Ryohichi Sugimura
University of Hong Kong
L3-64, Laboratory Block, 21 Sassoon Road, Hong Kong
Hong Kong

Dear Dr. Sugimura,

Thank you for submitting your Research Article entitled "PD-L1 regulates inflammatory programs of macrophages from human pluripotent stem cells". It is a pleasure to let you know that your manuscript is now accepted for publication in Life Science Alliance. Congratulations on this interesting work.

DISTRIBUTION OF MATERIALS:

Again, congratulations on a very nice paper. I hope you found the review process to be constructive and are pleased with how the manuscript was handled editorially. We look forward to future exciting submissions from your lab.

Sincerely,
